# Systematic Review of Recurrent Osteosarcoma Systemic Therapy

**DOI:** 10.3390/cancers13081757

**Published:** 2021-04-07

**Authors:** Ioanna Gazouli, Anastasios Kyriazoglou, Ioannis Kotsantis, Maria Anastasiou, Anastasios Pantazopoulos, Maria Prevezanou, Ioannis Chatzidakis, Georgios Kavourakis, Panagiota Economopoulou, Vasileios Kontogeorgakos, Panayiotis Papagelopoulos, Amanda Psyrri

**Affiliations:** 1Department of Medical Oncology, University Hospital of Ioannina, 45500 Ioannina, Greece; ioannagz@yahoo.gr; 2Second Propaedeutic Department of Medicine, Attikon University Hospital, 1 Rimini Street, Chaidari, 12462 Athens, Greece; ikotsantis@gmail.com (I.K.); miriamanastasiou9@gmail.com (M.A.); anastasios.pantazopoulos@gmail.com (A.P.); mariaprevezanou@gmail.com (M.P.); biovazelos@hotmail.com (I.C.); georgekavourakis@yahoo.gr (G.K.); panagiota_oiko@hotmail.com (P.E.); dpsyrri@med.uoa.gr (A.P.); 3First Department of Orthopaedic Surgery, Attikon University General Hospital, Chaidari, 12462 Athens, Greece; vkonto@med.uoa.gr (V.K.); pjportho@med.uoa.gr (P.P.)

**Keywords:** recurrent/refractory osteosarcoma, chemotherapy, systemic treatment, immunotherapy

## Abstract

**Simple Summary:**

Osteosarcoma is the most common primary bone cancer. Its therapeutic approach includes cytotoxic chemotherapy and surgery. However, when recurrence or metastasis occurs the therapeutic options are limited with poor results. Herein we have conducted a systematic review of the systemic treatment options in recurrent and/or metastatic osteosarcoma over the last two decades. Our results indicate the paucity of our therapeutic armamentarium for this entity, with the majority of the studied modalities resulting in limited or no benefits. Intense translational research and future clinical studies reveal the unmet need for new treatment options for osteosarcoma patients with metastatic and/or recurrent disease.

**Abstract:**

Osteosarcoma is the most frequent primary bone cancer, mainly affecting those of young ages. Although surgery combined with cytotoxic chemotherapy has significantly increased the chances of cure, recurrent and refractory disease still impose a tough therapeutic challenge. We performed a systematic literature review of the available clinical evidence, regarding treatment of recurrent and/or refractory osteosarcoma over the last two decades. Among the 72 eligible studies, there were 56 prospective clinical trials, primarily multicentric, single arm, phase I or II and non-randomized. Evaluated treatment strategies included cytotoxic chemotherapy, tyrosine kinase and mTOR inhibitors and other targeted agents, as well as immunotherapy and combinatorial approaches. Unfortunately, most treatments have failed to induce objective responses, albeit some of them may sustain disease control. No driver mutations have been recognized, to serve as effective treatment targets, and predictive biomarkers of potential treatment effectiveness are lacking. Hopefully, ongoing and future clinical and preclinical research will unlock the underlying biologic mechanisms of recurrent and refractory osteosarcoma, expanding the therapeutic choices available to pre-treated osteosarcoma patients.

## 1. Introduction

Osteosarcoma comprises 1% of all human malignancies. It is the most frequent primary bone cancer, usually arising in the metaphyses of long bones of children and adolescents [1,2,3]. In older patients, it may arise in the axial skeleton, often complicating Paget’s disease [4]. In fact, two incidence peaks have been reported, the first among 15–20 and the second among 75–80-year-old patients, with 8 and 6 newly diagnosed cases per million persons every year, respectively [1].

Osteosarcoma is primarily treated with surgical excision not only of the primary but also of secondary lesions, if feasible [5,6]. Nonetheless, surgery alone, as applied in the 1960s, resulted in disappointing overall survival rates of 11%. Within one year from treatment up to 90% of operated patients experienced disease relapse, most often with lung metastases. Fortunately, the initiation of adjuvant chemotherapy in the 1970s increased the 5-year overall survival rates up to 68% [2,7,8]. Unluckily, no further significant improvement has been made to date. Relapsed as well as primarily inoperable and/or refractory osteosarcoma, continue to be hardly manageable entities, with poor prognosis [9]. Characteristically, overall survival of recurrent osteosarcoma patients is reported to be lower than 28% at 5 years of relapse, in patients not amenable to metastasectomy [10,11,12].

In this context, we questioned what systematic treatment strategies have been developed against recurrent osteosarcoma, within the last two decades. Our main queries were: 1. Are there any treatment strategies able to initiate objective responses or disease control in recurrent osteosarcoma patients? 2. Is disease control durable, once achieved? 3. Is there any evidence of experimental treatments extending patient survival? 4. Which treatment strategies have been successfully combined? 5. Do novel approaches (targeted treatment, immunotherapy) have any significant impact on recurrent osteosarcoma treatment? 6. Are there any studies comparing the relative effectiveness of different potential treatments or currently treated patients to historical controls?


We attempted to answer the abovementioned questions, by performing the present systematic literature review, focusing on the available clinical evidence regarding the systematic treatment of recurrent osteosarcoma.


## 2. Methods

This study was conducted and reported in accordance with the Preferred Reporting Items for Systematic Reviews and Meta-Analyses (PRISMA) statement and the common practices in the field. Eligible articles were identified by a search of MEDLINE bibliographical database for the period of 1 January 2000 up to 31 December 2020. The search strategy included the following keywords: osteosarcoma AND (bone neoplasm OR cancer OR sarcoma) AND (recurrent OR refractory) AND (guidelines OR consensus OR practice OR recommendation OR trial OR study) AND (chemotherapy OR systemic therapy OR management). Two investigators (IG and AK), working independently, searched the literature and extracted data from each eligible study. Language restrictions were applied (only articles in English, French and German were considered eligible). Reviews, experts’ opinions and prospective and retrospective studies were included in the analysis, whilst case reports were excluded for this systematic review. Manuscripts that did not state the name of the authors were excluded. Additional articles were identified from the reference lists of retrieved articles.

## 3. Results

Our initial search resulted in 282 articles. We excluded four articles due to language restrictions. Furthermore, 12 case reports and/or case report series were excluded as well. Additionally, 211 additional articles focusing on osteosarcoma diagnosis, prognosis and biology, as well as articles presenting surgical approaches and primary or adjuvant osteosarcoma treatment or preclinical assays, were considered to be outside of the scope of the present review. Sixteen studies (12 prospective clinical trials, 4 retrospective studies) were recovered from the reference lists of rejected articles. Overall, 71 articles were considered eligible to be included in our final analysis (Figure 1).

We identified 56 prospective clinical trials, published between 2000 and 2020, among which 20 were phase I trials, 6 were phase I/II trials and 30 were phase II trials. No phase III trials with published results were found. All studies were single arm, with the exception of two recent, randomized, placebo-controlled trials [13,14]. Characteristically, most of these trials recruited patients with recurrent and/or refractory solid tumors or bone sarcomas, with solely 18 among them [13,14,15,16,17,18,19,20,21,22,23,24,25,26,27,28,29,30] exclusively recruiting recurrent and/or refractory osteosarcoma patients. Patient age range varied vastly, from 1 to 85 years of life, with about half of the included studies (27 out of 56), involving patients younger than 30 years. Objective responses were not achieved in most trials, while the highest objective response rate noted was 78% in relapsed/refractory osteosarcoma patients treated with high dose carboplatin plus etoposide, followed by stem cell transplantation [29]. The best partial response rate to be attained by single-agent treatment (43.2%) was noted in the phase II trial of apatinib, published in 2018 [18]. 

In addition, we identified two metanalyses [31,32] and three systematic reviews of clinical trials [33,34,35]; one systematic review also included retrospective studies [35]. Finally, 10 publications reporting retrospective clinical data were considered suitable for this review [36,37,38,39,40,41,42,43,44,45].

### 3.1. Chemotherapy

Table 1 summarizes the chemotherapy regimens tested against recurrent and/or refractory osteosarcoma (OST). The microtubule inhibitor eribulin, was recently tested in 19 patients with recurrent/refractory OST, without any objective response, while all patients had already progressed within 4 months of treatment [16]. The double regimen of gemcitabine and docetaxel has been found to induce a partial response rate of 17% and a disease control rate of 40%, in OST patients, in a phase II trial recruiting patients with refractory bone sarcomas [46]. Median PFS and OS in the overall study population were 3.5 and 7.5 months, respectively. The same combination, when assessed in an earlier trial [47], gave inferior results (PR in 1 of 14 OST patients). Gemcitabine, when combined with oxaliplatin, attained one partial response among 12 OST patients [48]. As monotherapy, gemcitabine has not been shown to induce tumor shrinkage although it achieved disease stabilization in 4 out of 6 OST heavily pretreated patients participating in the trial for 3–24 months [49]. 

Monotherapy with docetaxel has been found to achieve a disease control rate of 26% of OST patients, lasting up to 10 months [50]; in this trial 1 patient experienced complete response, another one partial response and 4 more patients had stable disease. Pemetrexed has been employed against refractory OST, but without attaining any objective responses [51], with stable disease in one of 10 included patients.

Dual treatment with pirarubicin and cisplatin has been assessed in a phase II trial recruiting refractory OST patients [23]; 13% of patients experienced partial response, while disease control rate was about 35%. Half of the trial population was still free of disease progression and alive at 2 and 10 months, respectively. Monotherapy with inhaled lipid cisplatin particles for refractory OST patients with lung metastases was examined in 2013 [22], inducing one partial response and stabilizing disease in another two patients with low bulk lung disease. Patients with bulky lung disease did not benefit from treatment, while 13 out of 19 patients experienced respiratory adverse events being as severe as grade 3 in one patient.

Trabectedin has not been found to induce tumor shrinkage [52] in refractory OST patients, but induced stable disease in three of them, lasting up to the fourth, sixth and eighteenth cycle of treatment, respectively. Similarly, vinorelbine combined with metronomic cyclophosphamide, induced stable disease in 1 out of 10 OST patients, without any objective response [53]. 

Furthermore, 19% of OST patients experienced partial response after two cycles of cyclophosphamide and etoposide, with a disease control rate up to 54% [26], and 46% of patients free of disease progression at 4 months of treatment enrollment. Combination of cyclophosphamide with topotecan also yielded a disease control rate of 39%, with partial response in 2 of 18 refractory OST patients [54]. On the contrary, single agent temozolomide [55] and topotecan [56] did not prove effective against recurrent/refractory OST. One out of 8 OST pediatric patients receiving metronomic oral etoposide [57] has presented a partial response, with complete disease regression after radiotherapy, being free of disease beyond 6 years after the original OST relapse; another 4 patients had stable disease, persistent for 2 to 10 months. 

High dose chemotherapy with carboplatin and etoposide, followed by stem cell transplant, may induce a complete response rate up to 78%, with a 3-year disease-free survival rate up to 12% [29].

### 3.2. Tyrosine Kinase Inhibitors (TKIs)

Regorafenib has been compared to a placebo, in recurrent/refractory osteosarcoma in two multicenter, randomized, phase II clinical trials, both of which met their primary endpoint [13,14]. In the French study by Duffaud et al., 38 OST patients were 2:1 randomized to receive either regorafenib or placebo, resulting in a median PFS of 16 (95%CI8.0–27.3) versus 4 (95%CI 3.0–5.7) weeks, respectively. Similarly, in the North American trial by Davis et al. [14], a median PFS of 3.6 months was achieved among the 22 patients treated with regorafenib, versus 1.7 months for the placebo group (HR 0.42; 95% CI 0.21–0.85, *p* = 0.017) albeit, no significant difference was noted in overall survival (11.1 and 13.4 months for the regorafenib and the placebo arm, respectively, *p* = 0.62). Notably, partial tumor responses to regorafenib were also observed in both trials, in 7.6% and 13.6% of regorafenib treated patients, in the French [13] and in the American trial [14], respectively.

Other multi-tyrosine kinase inhibitors (Table 2) have been tested in single arm studies. The antiangiogenic TKI cabozatinib, [60] when administered to OST patients pretreated with one or more treatment lines, was shown to induce partial responses in 12% and tumor shrinkage of any degree in 41% of patients, while inhibiting disease progression during 6 months in one third of the patients. Median PFS and OS reached 6.7 and 10.6 months, respectively. A higher partial response rate (43%) was noted in the single arm trial of apatinib, where about 58% and 37% of pretreated OST patients were free of disease progression at 4 and 6 months of treatment initiation, respectively [18]. Safety and tolerability of axitinib was investigated in a phase I trial of young patients with refractory solid neoplasms; both the two OST patients included were free of disease progression at 6 months of treatment, although no response was noted [61]. Partial response in 8.6% of OST patients has also been attained in a phase II trial of sorafenib [24], with 46% of the study population being free of disease progression at 4 months of treatment, and 9% still receiving treatment beyond 6 months. Imatinib [62] has also been tested, but it did not confer any objective benefit to refractory OST patients.

Co-administration of TKIs with cytotoxic agents has also been investigated in three clinical trials. Lenvatinib was generally well tolerated in young OST patients in combination with etoposide and ifosfamide, achieving a disease control rate of 50% [20], although it induced hypertension and cerebral ischemia in one case. The phase I trial of gefitinib and irinotecan combination [63] reported no benefit for OST patients, despite showing some activity in the other tumor types examined. Erlotinib and temozolomide was another tolerable double regimen, but it was not shown to be of any effectiveness against recurrent OST [64]. 

### 3.3. mTOR Inhibitors

mTOR inhibitors, alone or in combination with other targeted or chemotherapeutic agents, have been explored in phase I and II trials (Table 3). Everolimus was at first administered in 25 patients with refractory solid tumors, including only two patients with recurrent/refractory osteosarcoma, resulting in disease stabilization in 1 of them [65]. In a subsequent study [21], it was co-administered with sorafenib in 38 OST patients, inducing partial response in 5% of them and disease control in 63%, lasting from 1 up to 10.6 months. Nonetheless, the trial did not meet its primary endpoint of 6-month PFS rate of 50% or greater. 

Sirolimus was combined with the antimetabolite gemcitabine in a recent phase II trial [19] resulting in a partial response rate of 6% and stable disease rate of 42%, while 44% of patients were still free of disease progression at 4 months of treatment, meeting the predefined 4-month PFS rate cutoff of 40%, hence the trial was deemed positive. More recently, sirolimus was combined with the cyclooxygenase inhibitor celecoxib and metronomic oral etoposide and cyclophosphamide [66] in patients with refractory solid neoplasms, including only 1 OST patient, without any significant results, other than the safety of the combination. 

Another mTOR inhibitor, ridaforolimus, has been tested in a single arm trial recruiting patients with refractory sarcomas [67]. Although the number of osteosarcoma patients is not specified, partial response was noted in two out of 54 bone sarcoma patients, while overall disease control rate reached 31%, with a median PFS of 15.4 months. 

Combination of temsirolimus with cixutumumab, a monoclonal antibody targeting the receptor of the Insulin-like Growth Factor-1, has been investigated in two single arm studies. In the study of Schwartz et al. [68], 3 out of 24 OST patients experienced a partial response, with 50% of them still free of disease progression at 6 weeks of treatment. Characteristically, 63% of OST samples expressed IGF-1 receptor. In contrast, the study of Wagner et al. [69], did not report any benefit for the included recurrent OST patients. A three-agent regimen of temsirolimus, irinotecan and temozolomide was tolerable in pediatric and adolescent refractory solid tumor patients, but it did not confer any responses [70]. 

### 3.4. Immunotherapy

Five multicenter trials have employed immune checkpoint inhibitors against recurrent/refractory osteosarcoma (Table 4). Only pembrolizumab has been found to induce an objective response in 1 out of 22 patients, while it stabilized disease for 27% of them, regardless of zero PD-L1 expression in the examined specimens [71]. Nonetheless, the most recently published phase II trial of pembrolizumab in recurrent, advanced osteosarcoma patients, did not meet its primary endpoint of achieving disease control at 18 weeks of treatment, inducing solely one metabolic response in a patient with high PD-L1 expression (>50%) [15]. 

Nivolumab prevented disease progression in 46% of patients for about one month after treatment initiation, but this was sustained beyond the first month of treatment only in one out of 11 evaluable OST patients [72]. According to the recently announced results of the IMMUNOSARC trial, nivolumab may be safely combined with sunitinib, achieving durable disease control in 55% of refractory bone sarcoma patients, lasting 6 months and beyond. Among the 17 osteosarcoma patients participating in this trial, one had a partial response, lasting up to 5.7 months [73].

Ipilimumab was shown to achieve disease control in 25% of OST patients, but at a dose over the study’s MTD, causing grade 3 colitis in one out of the two patients that had stable disease at 6 weeks of treatment [74].

In a trial of 2015 [75], HER2-directed CART-cells, were administered in 16 OST patients, most of them with HER2 expressing tumours (HER2 expression 26% or higher in 15 of the patient study population), inducing SD in 3 OST patients, that persisted for 12 weeks. Impressively, in one patient, tumor necrosis of 90% was histologically discovered, after sequential surgical excision of his tumor, but this strategy was not reemployed in any later studies.

### 3.5. Targeted Agents-Miscellaneous

Multiple targeted agents have been also tested against recurrent/refractory osteosarcoma (Table 5). Most recently, the poly-ADP-ribose polymerase 1/2 inhibitor talazoparib has been safely combined with low dose temozolomide [76] in young patients with refractory solid tumors, but no disease control was achieved in OST patients. The monoclonal antibody glembatumumab vedotin (GV) targets a type I transmembrane glycoprotein, named glycoprotein non-metastatic B (gpNMB), which enhances cellular growth and metastatic potential, and has been found to be frequently overexpressed in osteosarcoma cells [77,78]. When administered in 22 OST patients, GV induced partial response in one patient and stable disease in another two, with a disease control duration of 4 months in all three patients [17]. All three patients strongly expressed gpNMB, but this was not related to disease control at 4 months (*p* = 0.68).

Targeting the Insulin-Like Growth Factor-1 Receptor (IGF-1R), a protein involved in the pathogenesis of bone and soft tissue sarcomas, is another strategy explored in osteosarcoma trials. The anti-IGF-1R antibody RG1507 (R1507) [83] protected 2 OST patients from disease progression beyond 13 and 20 months of treatment. In 2014 [80], it was re-administered in patients with refractory sarcomas, attaining partial responses in 5% of OST patients, with a disease control rate up to 31.6%. Overall median PFS was 5.7 (95% CI, 5.6–5.9) weeks and overall median OS reached 11 (95% CI, 9.4–13.1) months. Notably, 10 OST patients were alive beyond 2 years after study enrollment. Nevertheless, it was not further explored as its development was discontinued by the producer company [86]. Monotherapy with the above mentioned anti-IGF-1R antibody cixutumumab did not induce any responses among 11 OST patients enrolled in the trial, although it stabilized disease in one of them for 5 months [79]. 

Imetelstat is an oligonucleotide acting as a telomerase inhibitor that was well tolerated by pediatric and adolescent patients with refractory solid neoplasms, while inducing one partial response among the six recruited OST patients [81]; albeit treatment was interrupted because of severe thrombocytopenia. In this trial, telomerase inhibition was confirmed in peripheral blood mononuclear cells by specialized kits. 

Lexatumumab is a monoclonic anti-TNF-receptor 2 [82] that did not induce any objective responses in pediatric patients with refractory solid tumors. Nonetheless, one 16-year-old OST patient had a metabolic response, with resolution of her baseline hypermetabolic metastatic lung lesions in the PET/CT scan, and parallel lesion calcification in the CT imaging; the patient was reported to have a continuously negative PET/CT scan beyond one year off treatment. 

RexinG, is a cyclin G1 construct with cytotoxic activity. It was tested in 2009 [84] and managed to maintain stable disease in 59% of OST patients, with a median PFS beyond 3 months and a median OS of 7 months. Additionally, assessed by PET/CT, 4 out of 17 evaluable OST patients, presented with a persistent metabolic response. 

The 17-AAG (17-N-allylamino-17-demethoxygeldanamycin) inhibits the action of heat shock protein 90. When administered in pediatric patients with refractory OST, no responses were noted; importantly, there were two deaths during the first cycle of treatment in OST patients with bulky lung lesions [85].

Ecteinascidin 743, an alkaloid thought to interfere with the transcriptional process, was also tested in an early trial [28]. Objective responses of 49%, 36% and 25% were noted in 3 OST patients, with 24% of the OST patients having stable disease. Subsequent trials of this particular agent are lacking. 

Inhalation of GM-CSF in OST patients with lung metastatic lesions was not proven to exert any immunomodulatory effect, while the one-year event free survival rate of treated patients did not exceed 20% (95% CI: 10–34%) [25]. Radioactive samarium 153 may have a palliative effect in patients with refractory OST and bone metastatic lesions [27,30].

### 3.6. Metanalyses of Clinical Trials

Results of metanalyses of clinical studies are summarized in Table 6. A systematic review of phase II trials, investigating treatment of recurrent OST, published in 2017 [33], included 99 trials, including 19 with published results, by the time of the data extraction. Nine out of 99 trials were randomized, with 40, 26 and another 26 of them employing targeted agents with or without chemotherapy or immunotherapy, chemotherapy alone and immunotherapy with or without chemotherapy, respectively. Most trials were multicenter (65 out of 99), and most of them located in the USA. 17 trials were international. Only 7 of the 19 published trials specifically recruited OST patients, while the rest 12 included other refractory neoplasms. Furthermore, 390 evaluable patients participated in these 19 single arm trials, with only 3 of them experiencing complete response and 16 of them partial response, leading to an overall response rate of 5%; overall disease control rate was 33%, with an additional 109 patients (28%) experiencing stable disease after treatment initiation. Remarkably, the outcomes of only three trials met their satisfied their predefined endpoint [19,24,84].

A retrospective analysis of seven single arm, single agent phase II trials with an osteosarcoma cohort [31], run by the Children’s Oncology Group (COG), the Children’s Cancer Group and the Pediatric Oncology Group, between 1997 and 2007, resulted in an overall 4-month event free survival rate of 12% (6–19%) of the participating pediatric osteosarcoma patients. Among the 96 study participants, only 2 OST patients treated with single agent docetaxel in the trial by Zwerdling et al., [49] experienced an objective response (ORR 2%).

A metanalysis [35] of 2 prospective phase II trials [42,47] and another 2 retrospective studies [38,40], involving 66 patients with recurrent/refractory OST, yielded an overall response rate of 12%. 

A systematic review of 20 phase I/II trials, published between 1990 and 2013, employing six different chemotherapy regimens against recurrent/refractory OST, involving a total of 285 patients, performed in 2014 [34], indicates that the combination of Ifosfamide, Etoposide, High Dose Methotrexate achieved the highest objective response rate (62%). The second higher ORR (41%) has been noted among trials treating OST patients with ifosfamide and etoposide, while the combination of topotecan and cyclophosphamide induced the lowest ORR (12%). 

In accordance with the above findings, the combination of ifosfamide, etoposide and carboplatin, employed in three phase I/II trials of the Children’s Cancer Group, showed an overall response rate of 36% among the evaluable 34 refractory OST patients. Stable disease was attained in another 38% of OST patients, with a one-year survival rate up to 41% [32]. 

### 3.7. Retrospective Data

Clinical retrospective data are summarized in Table 7. A retrospective review of 28 pediatric and adolescent Chinese patients with refractory OST, treated with gemcitabine and docetaxel, showed an overall response rate up to 23.5% (3 complete responses and 1 partial response out of 17 evaluable patients), persisting for 11 months, with a disease control rate up to 41.2% [38]; retrospectively assessed median PFS and OS were 2 and 8 months, respectively.

Nonetheless, when gemcitabine-docetaxel combination was retrospectively compared to pirarubicin based regimens [40], as a salvage treatment, it attained lower objective response rates among OST patients (13 vs. 25%), as well as a significantly shorter median overall survival (14 vs. 9 months, *p* < 0.05). 

High dose thiotepa, followed by an autologous stem cell transplant, has been retrospectively found to induce a radiologic response in 31% of OST patients, with a median PFS of 8.8 months [39]. In another retrospective assessment of ASCT following high dose multi-agent chemotherapy regimens (melphalan-etoposide, melphalan-etoposide-carboplatin, carboplatin-etoposide-thiotepa, carboplatin-etoposide- cyclophosphamide), 2 persistent complete remissions were observed, among the 12 evaluable OST patients. Relapse free survival rate at 8 months of treatment was 20% and 16-month- overall survival rate was 29%. No notable benefit due to multi-agent chemotherapy was noted, while 3 out of 15 initial participants died because of severe toxicity [45]. In a more recent retrospective analysis of 19 pediatric patients with refractory OST, 15 among them succeeded complete remission after ASCT, persisting from 8.5 to 90 months [37].

## 4. Discussion

We performed a systematic literature review of the available clinical evidence regarding the treatment of recurrent osteosarcoma. For this purpose, we explored the available clinical evidence that has been published within the period 2000–2020. We came up with 56 eligible clinical trials, 10 retrospective studies and 5 reviews and metanalyses of clinical studies that were presented in this review. 

Current practice in the systematic treatment of recurrent osteosarcoma, is based on double or triple cytotoxic agent regimens (e.g., Ifosfamide, Etoposide and High Dose Methotrexate, ifosfamide or cyclophosphamide and etoposide, gemcitabine and docetaxel); in a relevant systematic review of 20 clinical trials [34], the overall objective response rate varies from 12% for cyclophosphamide plus topotecan, up to 42% for ifosfamide plus etoposide, rising up to 62% with the addition of high dose methotrexate to the latter, albeit with considerable hematologic toxicity. The combination of gemcitabine and docetaxel seems to attain more modest results, with overall response rates from 5.6 to 17%, in prospective and retrospective studies [34,38,42,43,46]. In accordance with the above study outcomes, the combination of ifosfamide and etoposide is the recommended treatment of osteosarcoma recurrences, while gemcitabine and docetaxel, cyclophosphamide and topotecan are secondary choices [87].

Although anthracyclines have not been largely employed against recurrent osteosarcoma, chemotherapy with pirarubicin, a more selective analogue of doxorubicin, has been shown to induce responses in up to 25% of patients, implying it should be further investigated in future clinical trials [23,40]. Of note, single agent chemotherapy treatment with topotecan, temozolomide, pemetrexed, eribulin, gemcitabine and trabectedin, has poor activity, with almost no objective responses and disease control in about 10% of treated patients overall [16,31,48,50,52,55,56].

High dose chemotherapy, followed by stem cell transplantation, has been investigated in one clinical trial, leading 78% of osteosarcoma patients to complete remission [29]. More recent evidence derives from retrospective studies including mostly pediatric and adolescent patients with recurrent and/or refractory osteosarcoma, inducing complete remission rates from 12 to 79% of osteosarcoma patients, during longer than 8 months [36,37,39,45]. Although it has not been investigated among older patients, it is a considerable treatment option for younger ages, whose value remains to be confirmed in prospective as well as randomized clinical trials. 

There is evidence that higher VEGF expression in osteosarcoma specimens significantly correlates with shorter disease free and overall survival of patients [88,89]. On this basis, various multi-tyrosine kinase inhibitors with antiangiogenic activity have been explored against recurrent osteosarcoma. In fact, regorafenib and sorafenib are recommended as second line treatment of recurrent osteosarcoma in the relevant international guidelines. Remarkably, regorafenib is the only agent that has been examined in the context of two randomized clinical trials [13,14], achieving partial responses in 7.6 and 13.6% of patients, and significantly prolonging the progression free survival, for about 3 months, compared to placebo. Sorafenib has shown similar results, with a disease control rate of 43% and median progression free survival at 4 months [24]. Cabozatinib is suggested as an alternative choice for recurrent osteosarcoma patients, inducing partial responses in 12% of patients, and a disease control rate of 33%, with a median PFS of 6 months and a median OS of 10 months in the CABONE trial [60]. Apatinib has achieved the highest partial response rate (43.2%), while controlling disease in all treated patients, with a PFS ranging between 3 and 7 months [18]. Nonetheless, none of the latter three TKIs has been evaluated in any comparative clinical studies to date. Remarkably, sunitinib has been co-administered with nivolumab in patients with refractory, advanced bone sarcomas, inducing durable disease control in 55% of them, as well as a partial response in one osteosarcoma patient. Hence, the combination of TKIs with immune checkpoint inhibitors, may be a promising choice, following the paradigm of soft tissue sarcomas [90,91].

mTOR inhibition has been proposed as an attractive therapeutic approach, given the mTOR/PI3K mutations promoting proliferation and metastasis while inhibiting apoptosis in osteosarcoma cell models [92,93]. Unfortunately, mTOR inhibitors have not proven their value in clinical trials so far (Table 3), but when combined with targeted or cytotoxic agents, they may exert some clinical activity. Sorafenib combined with everolimus [21] and sirolimus together with gemcitabine [19], may induce response in 5 and 6% of recurrent osteosarcoma patients and a disease control rate up to 63 and 48%, respectively, with the latter combination inducing more durable responses (4 vs.1 month). Currently, a novel mTOR inhibitor, LY3023414 is administered in pediatric patients carrying targetable mutations of the TSC (Tuberous sclerosis complex)/PI3K/mTOR pathways (NCT03213678). 

Since IGF-1R mediated signaling also seems to have an oncogenic effect in osteosarcomas [94], monoclonalanti-IGF-1R antibodies have been developed. Combination of the anti-IGF-1R monoclonal antibody cixutumumab with temsirolimus is tolerable while inducing a partial response rate of 13%, with a median PFS at 1.5 month and median overall survival above 7 months [68,69], but it has not yet been explored in further clinical trials. Another anti-IGF-1R monoclonal antibody, R1507 (RG1507), has shown encouraging results, inducing tumor shrinkage in 5% of OST patients, but its development has been terminated by the producer company. 

Miscellaneous targeted agents have also been tested against recurrent osteosarcoma, albeit with modest results. Glembatumumab vedotin (anti-glycoprotein non-metastatic B monoclonal antibody) and imetelstat (telomerase inhibitor oligonucleotide) may induce objective responses, in 4.5 and 17% of treated patients respectively, with imetelstat being administered in only 6 patients. Overall, stable disease rates noted in trials using the targeted treatments presented in Table 5, fluctuate between 9 and 24%, lasting around 2–4 months [17,28,79,80]. Notably, Rexin-G, a retroviral vector of a cytocidal cyclin construct, could be characterized as relatively promising, inducing a metabolic response in 24% of the study patients, and maintaining objective stable disease in 59% of them beyond 3 months [67]. 

The observation that osteosarcoma cells frequently express the receptor ERBB2 [95,96] has led to clinical trials assessing the effectiveness of HER2 blocking by trastuzumab (NCT04616560, NCT00005033), even though it did not have any notable effect as a primary metastatic osteosarcoma treatment [97]. Moreover, other monoclonal antibodies, targeting membrane proteins and glycolipids, such as pepinemab (NCT03320330) and dinutuximab (NCT02484443), initially intended to be used against central nervous tumors [98], are currently assessed also against recurrent and/or refractory osteosarcoma.

Interestingly, pediatric recurrent osteosarcoma patients can receive innovative targeted treatments, in the context of basket trials; selpercatinib (NCT04320888) is tested in patients carrying RET gene alterations, reported to be present in 4% of osteosarcomas [99]. Similarly, erdafitinib (NCT03210714) is assessed against refractory malignancies with targetable FGFR mutations, appearing in up to 18.5% of osteosarcomas and being associated with resistance to chemotherapy [100]. Disappointingly, NTRK fusions in osteosarcomas are infrequent and non-functional [101], implying osteosarcoma patients may not be prone to respond to the evolving NTRK targeted agents, vastly explored against other solid tumors. 

Although knock-down of PARP1 in osteosarcoma cells in vitro reduced cellular growth and enhanced apoptosis as well as susceptibility to chemotherapy [102], the PARP1 inhibitor talazoparib did not show any significant clinical activity in a recent trial [76]. Nonetheless, PARP inhibitors are currently investigated in three phase single arm clinical trials, treating patients with olaparib plus ceralasertib (ataxia telangiectasia and rad3 related kinase inhibitor) (NCT04417062), palbociclib (NCT03526250) and single agent olaparib (NCT03233204).

Unlike non-immunogenic soft tissue sarcomas, osteosarcomas are more susceptible to immunotherapy, due to their higher mutational burden and more prominent cytotoxic T-lymphocyte infiltration [103]. Indeed, there is evidence of the effectiveness of PD-1/PD-L1 inhibition in osteosarcoma murine models [104]. In the clinical setting, immune checkpoint inhibitors seem able to prevent disease progression in recurrent osteosarcoma patients, at rates from 26 to 46%, for 1–2 months, although they do not seem to induce any objective responses. Recently, pembrolizumab did not attain any objective responses among recurrent osteosarcoma patients, although four out of six patients evaluated with PET/CT had a metabolic response at six weeks of treatment. Notably, high PD-L1 expression was found only in 1 of the 11 evaluable patients (>50%), who experienced a mixed metabolic response, indicating that TPS may not be an accurate predictive biomarker in osteosarcoma treatment [15]. Given that single agent ipilimumab is only effective at high doses, exerting considerable toxicity, the well-established combination of nivolumab and ipilimumab is currently tested (NCT02304458). In another trial nivolumab is combined with the antimetabolite azacytidine (NCT03628209). Single agent avelumab (NCT03006848) and the combination of oleclumab (anti-CD37monoclonal antibody) and durvalumab (NCT04668300) are also under investigation. Administration of autologous tumor infiltrating cells in patients with refractory sarcomas and other solid tumors is an alternative immunotherapeutic approach, evaluated in recurrent osteosarcoma patients (NCT03449108). 

While performing this review, we observed a notable lack of clinical evidence concerning treatment of recurrent osteosarcoma, especially beyond the second line of chemotherapy. We also noticed a considerable lack of comparative, randomized clinical trials, attempting to compare the more innovative treatments against established choices or placebo, both among the already published studies, and among the ongoing ones. In contrast to plenty other malignancies such as melanoma, lung and colorectal cancer, where the identification of targetable mutations has changed the disease course of many patients [105], we have not yet recognized such crucial mutations or chromosomal alterations driving osteosarcoma growth, relapse and chemo-resistance, which would allow us to develop highly effective treatment strategies. In parallel, the immune checkpoint inhibitors that have revolutionized clinical oncology during the last two decades, have not shown yet impressive results against recurrent osteosarcoma, while the predictive value of widely applied biomarkers, such as PD-L1 expression, must be clarified. Basket clinical trials may broaden patients’ therapeutic opportunities, enabling them to receive state-of-the-art treatment. Systematic tumor molecular profiling practices, feasible by the currently available technology, might help us unravel the underlying biology of recurrent osteosarcoma, in order to distinguish the most critical molecular mechanisms that could serve as treatment targets. 

Hopefully, in the era of personalized medicine, patients suffering from recurrent osteosarcoma or other infrequent malignancies may aspire to a better future. Innovative technologies such as Next Generation Sequencing, recognition of cancer and immune molecular signatures, methylation arrays and real time mutational profile assessment by liquid biopsies, promise to lay a foundation for patient-tailored treatments with clinically meaningful results. 

## 5. Conclusions

In conclusion, despite its relative rarity, osteosarcoma is a significant neoplasm, as it primarily affects young patients, and its recurrences still impose a tough therapeutic challenge. Future studies will hopefully shed light on its underlying pathophysiology, providing patients and physicians with more treatment choices. 

## Figures and Tables

**Figure 1 cancers-13-01757-f001:**
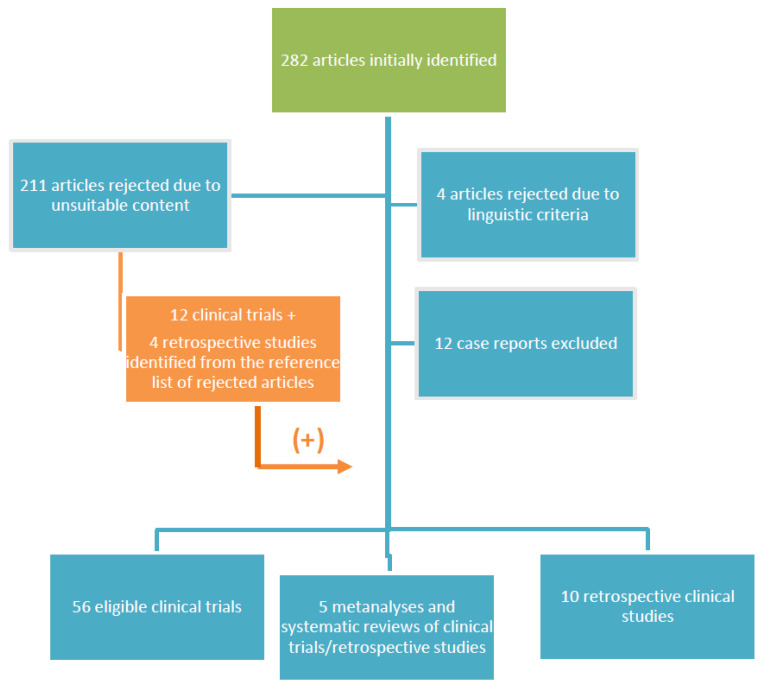
Diagram summarizing the elimination process of the initially identified published papers. There were 4 articles not written in English, 12 case reports and 211 articles with unsuitable content were excluded. Then, 16 eligible articles were recovered from the reference lists of the retrieved articles. Then, 56 clinical trials, 5 metanalyses and systematic reviews and 10 retrospective clinical studies were eventually included in the present review.

**Table 1 cancers-13-01757-t001:** Chemotherapeutic agents (19 trials).

Treatment	Ph	N	Age Range (Years)	OST pts	PR	SD	DC Duration	Outcomes	Reference
Eribulin	II	19	12–25	19	0	0	<4 mo	mPFS 38 days	Isakoff et al., 2019 [16]
Gemcitabine+ docetaxel	II	46	8–71	40	6/35 (17%)	14/35 (40%)	2–16 mo	Overall mPFS 3.5 mo (1–13.5) mOS 7.5 mo (2–45)	Palmerini et al., 2016 [46]
Inhaled cisplatin	I/II	19	13–27	19	1/19 (5%)	2/19 (10%)	Up to 12 mo	Grade 3 Respiratory toxicity in 1pt	Chou et al., 2013 [22]
Pemetrexed	II	72	3–23	10	0	1/10 (10%)	15 wks	-	Warwick et al., 2013 [51]
Pirarubicin + cisplatin	II	23	10–52	23	3/23 (13%)	5/23 (22%)	NS	mPFS 2 mo (2–11) mOS 10 mo (6–23)	Qi WX et al., 2012 [23]
Gemcitabine + Docetaxel	II	53	≥4	14	1/14 (7.1%)	NS	NS	-	Fox et al., 2012 [47]
vinorelbine + cyclophosphamide	II	117	1–24	10	0	1/10	NS	-	Minard-Colin et al., 2012 [53]
Trabectedin	I	12	8–16	3	0	3/3	12, 18 and 54 wks	-	Chuk et al., 2012 [52]
Gemcitabine + oxaliplatin	II	93	1–21	12	1/12 (8%)	4/12 (32%)	8 weeks	Overall mPFS 1.9 mo (1.77–2.2)	Geoerger et al., 2011 [48]
Cyclophosphamide + VP-16	II	26	8–47	26	5/26 (19%)	9/26 (35%)	NS	4 mo PFS 42% 4 mo OS 93% 1 yr OS 50%	Berger et al., 2009 [26]
weekly oxaliplatin	I	45	2–20	8	1/8 (12.5%)	NS	NS	Overall mPFS 1.8 mo	Geoerger et al., 2008 [58]
Docetaxel	II	177	1–27	23	1/23 (4%) CR: 1/23 (4%)	4/23 (16%)	NS/overall 12–42 wks	Overall 1yr OS 24%	Zwerdling et al., 2006 [50]
Topotecan	II	55	2–23	11	0	1/11	6 wks	overall 1yr OS 26% (13–39%)	Hawkins et al., 2006 [56]
Temozolomide	I	52	3–20	1	0	0	-	OST pt dies at 1st cycle of disease progression	De Sio et al., 2006 [55]
Metronomic oral VP16	II	21	3–16	8	1/8 (12.5%)	4/8 (50%)	2–10 mo	The 1 responding patient had a CR and remains NED at 79 mo of study entry	Kebudi et al., 2004 [57]
HD carboplatin + VP16 and ASCT	II	32		32	CR: 25/32 (78%)	0		mOS 23 months 3-year OS rate 20% 3-year DFS rate 12%	Fagioli et al., 2002 [29]
Cyclophosphamide + Topotecan	II	91	1–21	18	2/18 (11%)	5/18 (28%)	7–10 mo	-	Saylors et al., 2001 [54]
Gemcitabine	II	18	15–62	6	0	4/6	13–96 wks		Merimsky et al., 2000 [49]
Topotecan + HD cyclophosphamide	I	28	2–33	1	1	0	>2 mo	-	Kushner et al., 2000 [59]

Abbreviations: ph: Phase, OST: osteosarcoma, PR: partial response, SD: stable disease, DC duration: disease control duration, CR: Complete response, mo: months, wks: weeks, HD: High dose, VP-16:Etoposide, pt(s): patient(s), mPFS: median progression free survival, mOS: median overall survival, NS: Not Specified.

**Table 2 cancers-13-01757-t002:** Tyrosine Kinase Inhibitors (10 trials).

Treatment	Ph	N	Age Range (Years)	OST pts	PR	SD	DC Duration	Outcomes	Reference
Cabozatinib	II	90	20–53	45	5/42 (12%)	14/42 (33%)	6 mo	For OST pts: mPFS: 6.7 mo (5.4–7.9) mOS: 10.6 mo (7.4–12.5)	Italiano et al., 2020 [60]
Regorafenib vs. placebo	II	38	22–50	38	2/26 (7.6%) vs. none in placebo arm	17/26 (65%) vs. none in placebo arm	6–13 wks	mPFS: 16 wks (8.0–27.3) vs. 4 wks (3.0–5.7) in placebo arm	Duffaud et al., 2019 [13]
Regorafenib vs. placebo	II	42	18–76	42	3/22 (13.6%) vs. none in placebo arm	NS	NS	mPFS: 3.6 mo (2.0–7.6) vs. 1.7 mo (1.2–1.8) in placebo arm, HR 0.42 (0.21–0.85, *p* = 0.017) No difference in OS	Davis et al., 2019 [14]
Apatinib	II	37	16–62	37	16/37 (43.2%)	21/37 (56.8%)	4 mo	mPFS: 4.5 mo (3.5–6.2) mOS: 9.9 mo (8–18.9)	Xie et al., 2018 [18]
Axitinib	I	18	5–17	2	0	2/2 (100%)	6 mo	MTD 2.4 mg/m^2^/dose	Geller et al., 2018 [61]
Lenvatinib+ IFO + VP-16	I/II	16	2–25	16	1/16	7/16	NS	Ongoing trial	Gaspar et al., 2017 [20]
Gefitinib + IRN	I	16	5–21	5	NS	NS	NS	Tolerable in pediatric pts but cerebral ischemia in 1 patient	Brennan et al., 2014 [63]
Sorafenib	II	35	15–62	35	3/35 (8.6%)	12/35 (34%)	4 mo 3/35 (9%) of pts progression free beyond 6 mo	4 mo PFS: 46% (28–63%) mPFS: 4 mo (2–5), mOS: 7 mo (7–8)	Grignani et al., 2012 [24]
Imatinib	II	70	3–29	12	0	NS	NS	cKIT expression not tested	Bond et al., 2008 [62]
Erlotinib + TMZ	I	46	3–20	6	0	0	-	-	Jakacki et al., 2008 [64]

Abbreviations: IFO: Ifosfamide, TMZ: Temozolomide, IRN: Irinotecan, MTD: Maximal Tolerated dose.

**Table 3 cancers-13-01757-t003:** mTOR inhibitors (8 trials).

Treatment	Ph	N	Age Range (Years)	OST pts	PR	SD	DC Duration	Comments/Outcomes	Reference
sirolimus + celecoxib + metronomic VP-16 + Cyclophosphamide	I	18	2–19	1	0	NS	-	Sirolimus MTD 2 mg/m^2^	Qayed et al., 2020 [66]
Sirolimus + Gemcitabine	II	33	3–60	33	2/33 (6%)	14/33 (42%)	4 mo	mPFS: 2.3 mo (0–5.2) mOS: 7.1 mo (2.8–11.4)	Martin-Broto et al., 2017 [19]
Everolimus + Sorafenib	II	38	18–64	38	2/38 (5%)	22/38 (58%)	1 mo	6 mo PFS rate 29% mPFS 5mo	Grignani et al., 2015 [21]
temsirolimus+ IRN + TMZ	I	71	1–22	7	0	0	-	Tolerable in pediatric patients	Bagatell et al., 2014 [70]
Temsirolimus + cixutumumab	I	46	1–27	11	0	NS	NS	IGF-1R expressed in 80% of OSTs mTOR expressed in 20% of OSTs	Wagner at al., 2014 [69]
Temsirolimus + cixutumumab	II	162	19–82	24	3/24 (13%)	NS	NS	mPFS 6 wks (6–15.7) mOS above 7.6 mo IGF-1R expressed in 63% of OSTs	Schwartz et al., 2013 [68]
Ridaforolimus	II	212	16–78	NS (54 bone sarcomas)	2	NS	NS	DCR 31% in bone sarcomas mPFS 15.4 wks	Chawla et al., 2012 [67]
Everolimus	I	25	3–21	2	0	1/2	8 mo	MTD 5 mg/m^2^	Fouladi et al., 2007 [65]

VP-16: etoposide, IRN: Irinotecan, TMZ: Temozolomide, DCR: Disease Control Rate.

**Table 4 cancers-13-01757-t004:** Trials employing immunotherapy (6 trials).

Treatment	Ph	N	Age Range (Years)	OST pts	PR	SD	DC Duration	Outcomes	Reference
pembrolizumab	II	12	19–55	12	0 (4/12 metabolic response in PET/CT at 6 weeks)	1/12 (8%)	9 wks	mPFS 1.7 mo (1.2–2.2) mOS 6.6 mon (3.8–9.3)	Boye et al., 2021 [15]
Nivolumab	I/II	85	8–17	13	0	5/11 (46%)	6 wks	TPS 1–3% in 2 OST pts	Davis, Fox et al., 2020 [72]
						1/11 (9%)	8 wks		
Nivolumab + sunitinib	I/II	40	21–74	17	1/17 PR (6%)	NS Overall 22/40 (55%)	6 mo and beyond at 45% of cases	Overall mPFS: 3.7 mo (3.4–4) mOS: 14.2 mo (7.1–21.3)	Palmerini et al., 2020 [73]
Pembrolizumab	II	84	16–70	22	1/22 (5%)	6/22 (27%)	NS (overall 8wks)	TPS 0% in OST pts	Tawbi et al., 2017 [71]
Ipilimumab	I	33	2–21	8	0	2/8 (25%) at 10 mg/kg	6 wks	MTD 5 mg/kg	Merchant et al., 2016 [74]
HER-2 specific CART-cells	I/II	19	7.7–29.6	16	0	3/16 (19%)	12–15 wks	HER2 ≥ 26% in 15/19 pts	Ahmed et al., 2015 [75]

Note: Objective response and SD assessments refer only to (OST)osteosarcoma patients, Ph: Phase, N study popylation, PR: partial response, SD: stable disease, DC duration: disease control duration, TPS: Tumor Proportion Score (PD-L1), MTD: maximal tolerated dose.

**Table 5 cancers-13-01757-t005:** Targeted agents—Miscellaneous (10 trials).

Treatment Regimen	Ph	N	Age Range (Years)	OST pts	PR	SD	DC Duration	Comments	Reference
Talazoparib + TMZ	I	40	4–25	4	0	0	-	-	Schafer et al., 2020 [76]
glembatumumab vedotin (GV)	II	22	12–31	22	1/22 (4.5%)	2/22 (9%)	4 mo	Strong gpNMB expression in 15/21 (68%) of pts	Kopp et al., 2019 [17]
Cixutumumab monotherapy	II	102	2–30	11	0	1/11 (9%)	5 mo	-	Weigel et al., 2014 [79]
R1507	II	163	7–85	38	2/38 (5%)	10/38 (26%)	NS	Overall mPFS: 5.7 wks (5.6–5.9) mOS: 11 (9.4–13.1)	Pappo et al., 2014 [80]
Imetelstat (oligonucleotide)	I	20	3–21	6	1/6 (17%)	NS	NS	Telomerase inhibition confirmed by Western Blot	Thompson et al., 2013 [81]
Lexatumumab	I	24	2–21	9	0	NS	24 mo	metabolic response in 1/9 OST pt (11%)	Merchant et al., 2012 [82]
RG1507	I	31	3–17	3	0	2/3 (66%)	Beyond 52 and 78 wks	-	Bagatell et al., 2011 [83]
Rexin-G	I/II	42	7–68	22	0	10/17 (59%)	≥3 mo	metabolic response in 4/17 (24%) of OST pts mPFS ≥3 mo mOS 6.9 mo	Chawla et al., 2009 [84]
17-AAG	I	15	4–21	6	0	0	-	2 OST pts with lung metastases died at cy 1	Bagatell et al., 2007 [85]
ecteinascidin 743	II	25	12–67	25	0	6/23 (24%)	2–4 mo	Minor response in 3/23 (12%) OST pts	Laverdiere et al., 2003 [28]

**Table 6 cancers-13-01757-t006:** Metanalyses (MET)/systematic reviews (SR) (five studies).

Treatment	Article Type	Trials Included	Total OST pts	Outcome of OST Patients		Reference
targeted therapy ± IO/CT: 40 trials	SR	99 ph II trials (9 randomised) 19 published trials	390	ORR 5% SD 28% DCR 33%		Omer et al., 2017 [33]
				4 month PFS: 10–55% 6 month PFS: 7–45%		
CT alone: 26 trials						
IO ± CT: 26 trials						
Docetaxel Topotecan irinotecan Rebeccamycin Imatinib Oxaliplatin Ixabepilone Aerosolized GM-CSF	MET	7 ph II trials	96	ORR: 2%		Lagmay et al., 2016 [31]
				4 mo EFS 12% (6–19%)		
IEM, IE, CE, ICE, CT, GEMDOX	SR	20 ph I/II trials	285	IEM	ORR: 62% DCR: 92.3%	Xiao et al., 2014 [34]
				IE	ORR 41.7% DCR 77.9%	
				CE	ORR 20.5 DCR 56.8 %	
				ICE	ORR 30% DCR 73.5%	
				CT	ORR 12% DCR 40%	
				GEMDOX	ORR 14.5% DCR 36.4%	
Gemcitabine based regimens	SR	2 phII trials 2 retrospective studies	66	pooled ORR = 12.1% (8/66)		Wei et al., 2014 [35]
ICE	MET	3 ph I/II	34	CR 18%, PR 18%, ORR 36%, SD 38%		Van Winkle et al., 2005 [32]
				1yr OS rate 41% 2yr OS rate 26%		

Note: CT chemotherapy, IO Immunotherapy, IEM: Ifosfamide, Etoposide, High Dose Methotrexate, IE: ifosfamide, etoposide, CE: Cyclophosphamide, Etoposide, ICE: ifosfamide, carboplatin, etoposide, CT: cyclophosphamide, topotecan, GEMDOX: gemcitabine and docetaxel, Ixabepilone: microtubule stabilizing agent.

**Table 7 cancers-13-01757-t007:** Retrospective data (10 studies).

Treatment	N	Age Range (Years)	OST pts	PR	SD	Comments	Reference
HD chemotherapy and ASCT	28	1–22	6	CR: 3/6 (50%)	0	Remission duration: 17, 18, 29 months after ASCT	Choi et al., 2016 [36]
HD chemotherapy and ASCT	19	5–16	19	CR: 15/19	0	OS 78.3% EFS 67.4% median follow-up of 31 mo (1–91)	Hong et al., 2015 [37]
Docetaxel + gemcitabine	28	5–20	28	1/17 (6%) CR: 3/17 (17.5%) ORR: 4/17 (23.5%)	3/17 (17.5%)	Median response duration 11.2 mo (2.8–14.6) 1-yr-OS rate 35.3 ± 11.6%	Song et al, 2014 [38]
HD thiotepa and ASCT	53	4–31	53	12/39 (31%)	18/39 (46%)	mPFS: 8.8 mo(2.3–15.3) mOS: 21.2 mo(8.9–13.4)	Marec-Berard et al., 2014 [39]
Pirarubicin based chemotherapy	52	7–59	52	13/52 (25%)	23/52 (44.2%)	Longer mOS with pirarubicin-based chemotherapy vs. gemcitabine-docetaxel treatment: (14.0 vs. 9.0 months, *p* < 0.05)	He, Qi et al., 2013 [40]
gemtabine + docetaxel	23		23	13% (3/23)	8/23 (34.8%)		
Gemcitabine + docetaxel	19	1–22	6	0	2/6	SD duration for 3 and 9 months in 2 OST pts	Rapkin et al., 2012 [41]
salvage gemcitabine + docetaxel	18	12–57	18	1/18 (5.6%)	3/18 (17%)	mPFS 2 mo (2–6) mOS: 8 mo (3–21)	Qi et al., 2012 [42]
Gemcitabine + docetaxel	22	8–23	17	3/17 (17.6%)	1/17 (6%)	Overall Median response duration 3.5 mo	Navid et al., 2008 [43]
Autologous stem cell transplant	36	2–26	1	0	0	OST patient died	Fraser et al., 2006 [44]
HD chemotherapy and ASCT	15	6–21	15	CR: 2/12 (17%)	2/12 (17%)	3/15 pts died of toxicity 2/12 pts relapse free at 9 mo 5/12 pts alive at 9 mo	Sauerbrey et al., 2001 [45]

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
