# Peer review of "Systematic Review of Recurrent Osteosarcoma Systemic Therapy"

_cancers, 2021, doi:10.3390/cancers13081757_

Round 1

Reviewer 1 Report

The manuscript entitles "Systematic review of recurrent osteosarcoma systemic therapy" by Ioanna  Gazouli et al. is a very interesting and comprehensive overview of the current therapeutic strategies available or under investigation to treat recurrent OS. 
The article is convincing and very well written.  The reviewer has only one comment to this complete review which is how could prersonalized medicine approaches and technologies help the development of secure and effective therapies?

Author Response

Hopefully, in the era of personalized medicine, patients suffering from recurrent osteosarcoma or other infrequent malignancies may aspire to a better future.  Innovative technologies such as Next Generation Sequencing, recognition of cancer and immune molecular signatures, methylation arrays and real time mutational profile assessment by liquid biopsies, promise to lay a foundation for patient-tailored treatments with clinically meaningful results.

Reviewer 2 Report

In their manuscript entitled “Systematic review of recurrent osteosarcoma systemic therapy”, Gazouli and colleagues review the recent papers which reported systemic treatment options in recurrent and/or metastatic osteosarcoma.  Osteosarcoma is a malignant tumor that is extremely difficult to treat for recurrence and research for its determination of truly appropriate treatment strategies is very important.  From that point of view, this review is very informative and covers the recent literature widely, the conclusion is adequate.

Author Response

Thank you for considering our paper and for your valuable comments.

Reviewer 3 Report

A comprehensive review of status quo and potential treatments.

OF course would like NEW and innovative treatments to work but this shows just how hard it is to do this.

Author Response

Thank you for your intriguing comment.

The discovery of applicable, novel treatments seems to be, indeed, very demanding-hopefully, future studies will try to satisfy these expectations.